# The *C. elegans* TspanC8 tetraspanin TSP-14 exhibits isoform-specific localization and function

Zhiyu Liu, Herong Shi[†], Jun Liu *

Department of Molecular Biology and Genetics, Cornell University, Ithaca, New York, United States of America

† Deceased.
* JL53@cornell.edu

**Data Availability Statement:** All relevant data are within the manuscript and its Supporting Information files.

**Funding:** This work was supported by NIH R01 GM103869 and R35 GM130351 to JL. Some

## Abstract

Tetraspanin proteins are a unique family of highly conserved four-pass transmembrane proteins in metazoans. While much is known about their biochemical properties, the in vivo functions and distribution patterns of different tetraspanin proteins are less understood. Previous studies have shown that two paralogous tetraspanins that belong to the TspanC8 subfamily, TSP-12 and TSP-14, function redundantly to promote both Notch signaling and bone morphogenetic protein (BMP) signaling in *C. elegans*. TSP-14 has two isoforms, TSP-14A and TSP-14B, where TSP-14B has an additional 24 amino acids at its N-terminus compared to TSP-14A. By generating isoform specific knock-ins and knock-outs using CRISPR, we found that TSP-14A and TSP-14B share distinct as well as overlapping expression patterns and functions. While TSP-14A functions redundantly with TSP-12 to regulate body size and embryonic and vulva development, TSP-14B primarily functions redundantly with TSP-12 to regulate postembryonic mesoderm development. Importantly, TSP-14A and TSP-14B exhibit distinct subcellular localization patterns. TSP-14A is localized apically and on early and late endosomes. TSP-14B is localized to the basolateral cell membrane. We further identified a di-leucine motif within the N-terminal 24 amino acids of TSP-14B that serves as a basolateral membrane targeting sequence, and showed that the basolateral membrane localization of TSP-14B is important for its function. Our work highlights the diverse and intricate functions of TspanC8 tetraspanins in *C. elegans*, and demonstrates the importance of dissecting the functions of these important proteins in an intact living organism.

## Author summary

Tetraspanin proteins are a unique family of highly conserved four-pass transmembrane proteins in higher eukaryotes. Abnormal expression of certain tetraspanins is associated with various types of diseases, including cancer. Understanding the functions of different tetraspanin proteins in vivo is crucial in deciphering the link between tetraspanins and their associated disease states. We have previously identified two tetraspanins, TSP-12 and TSP-14, that share redundant functions in regulating multiple aspects of *C. elegans*

strains were obtained from the C. elegans Genetics Center, which is funded by NIH Office of 27 Research Infrastructure Programs (P40 OD010440). The confocal imaging data was acquired through the Cornell University Biotechnology Resource Center, with NYSTEM (CO29155) and NIH (S10OD018516) funding for the shared Zeiss LSM880 confocal/multiphoton microscope. The funders had no role in study design, data collection and analysis, decision to publish, or preparation of the manuscript.

**Competing interests:** Author Herong Shi was unable to confirm her authorship contributions. On her behalf, the corresponding author has reported her contributions to the best of their knowledge. The authors have declared that no competing interests exist.

development. Here we show that TSP-14 has two protein isoforms. Using CRISPR knock-in and knock-out technology, we have found that the two isoforms share unique, as well as overlapping expression patterns and functions. Furthermore, they exhibit distinct subcellular localization patterns. Our work highlights the diverse and intricate functions of tetraspanin proteins in a living multicellular organism, and demonstrates that protein isoforms are another mechanism *C. elegans* uses to increase the diversity and versatility of its proteome.

## Introduction

Tetraspanin proteins are a unique family of four-pass transmembrane proteins that are highly conserved in metazoans [1,2]. There are 33 tetraspanins in humans, 37 in *Drosophila* and 21 in *C. elegans*. All tetraspanin proteins contain four transmembrane domains and two extracellular (EC) loops, including the small EC1 loop and the large EC2 loop. The large EC2 loop contains a conserved Cys-Cys-Gly (CCG) motif and additional cysteine residues that mediate disulfide bond formation. Based on the number of cysteine residues in the EC2 loop, tetraspanins can be classified into distinct subfamilies. One of them is the TspanC8 subfamily. There are six members of TspanC8 tetraspanins in mammals, Tspan5, Tspan10, Tspan14. Tspan15, Tspan17 and Tspan33 [3,4]. TspanC8 tetraspanins are known to bind directly to ADAM10 (A Disintegrin and Metalloprotease 10) to regulate its trafficking and function [5–7]. Evidence from recent studies showed that different TspanC8s localize ADAM10 to different subcellular compartments and differentially affect the ability of ADAM10 to cleave distinct substrates [8–12]. However, the large number of TspanC8 proteins in mammals makes it technically challenging to comprehensively decipher the endogenous localization and functions of each of these TspanC8 tetraspanins in live animals during development.

With the availability of molecular genetic tools to manipulate its genome and the possibility to perform high resolution imaging on live animals, *C. elegans* provides a unique model system for dissectting the functions of TspanC8 tetraspanin proteins in vivo. Unlike mammals, *C. elegans* has only one TspanC8 tetraspanin, TSP-14, which has a paralog TSP-12. Despite being a C6 tetraspanin, TSP-12 functions redundantly with TSP-14 to promote both Notch signaling and BMP signaling [13–15]. Specifically, while animals lacking either TSP-12 or TSP-14 do not exhibit any overt phenotypes, animals lacking both TSP-12 and TSP-14 are small (Sma), vulvaless (Vul), exhibit maternal effect embryonic lethality (EMB), and have suppression of the *sma-9(0)* coelomocyte defects (Susm) [16]. The Sma and Susm phenotypes of *tsp-12(0); tsp-14(0)* double mutants are due to defects in BMP signaling, a pathway known to regulate body size and postembryonic mesoderm development in *C. elegans* [17]. We have previously shown that endogenous TSP-12 and TSP-14 are both localized to the plasma membrane and to various intracellular vesicles that include early, late and recycling endosomes [16]. In the early embryo, TSP-12 is required for the cell surface localization of the *C. elegans* ADAM10 ortholog SUP-17 [15]. In hypodermal and intestinal cells in the developing larvae, TSP-12 and TSP-14 function redundantly to regulate the recycling of the type II BMP receptor DAF-4 to the cell surface [16].

During the course of these studies, we noticed that the *tsp-14* locus encodes two major protein isoforms, TSP-14A and TSP-14B, that differ by only 24 amino acids at the N-termini. Using a combination of CRISPR knock-in/knock-out technology and high-resolution microscopy, we assessed the endogenous expression, localization, and function of the two different TSP-14 isoforms in live animals. We show here that TSP-14A and TSP-14B share unique and

overlapping functions in *C. elegans* development. Moreover, while TSP-14A is localized to apical, intracellular vesicles, TSP-14B is localized to the plasma membrane on the basolateral side. We further identified the basolateral membrane localization signal within the 24 amino acids unique to TSP-14B, and showed that this sequence is essential for both the localization and function of TSP-14B. Our work highlights the diverse and intricate functions of TspanC8 tetraspanins in an intact living organism.

## Results

### The *tsp-14* locus produces two protein isoforms, TSP-14A and TSP-14B, which differ by 24 amino acids at their N-termini

Sequencing data from existing cDNA clones showed that the *tsp-14* locus produces two major TSP-14 transcripts, *tsp-14a* and *tsp-14b*, which share the same 3' ends, but differ at their 5' ends (wormbase.org). Specifically, *tsp-14b* uses an upstream alternative 35bp first exon. While the second exon of *tsp-14b* and the first exon of *tsp-14a* share the same 3' ends, the *tsp-14b* transcript has an additional 163bp segment upstream of the shared 126bp sequences (Fig 1A). Thus the *tsp-14* locus is predicted to produce two protein isoforms, TSP-14A and TSP-14B, which differ at their N-termini, with TSP-14B having an extra 24 amino acids at its N-terminus compared to TSP-14A (Fig 1B–1C). Both isoforms share the same four transmembrane domains, two extracellular loops that include eight conserved cysteine residues in the second extracellular loop, and an intracellular C-terminal tail. Whether there is any functional significance regarding the presence of the two TSP-14 isoforms was not yet known.

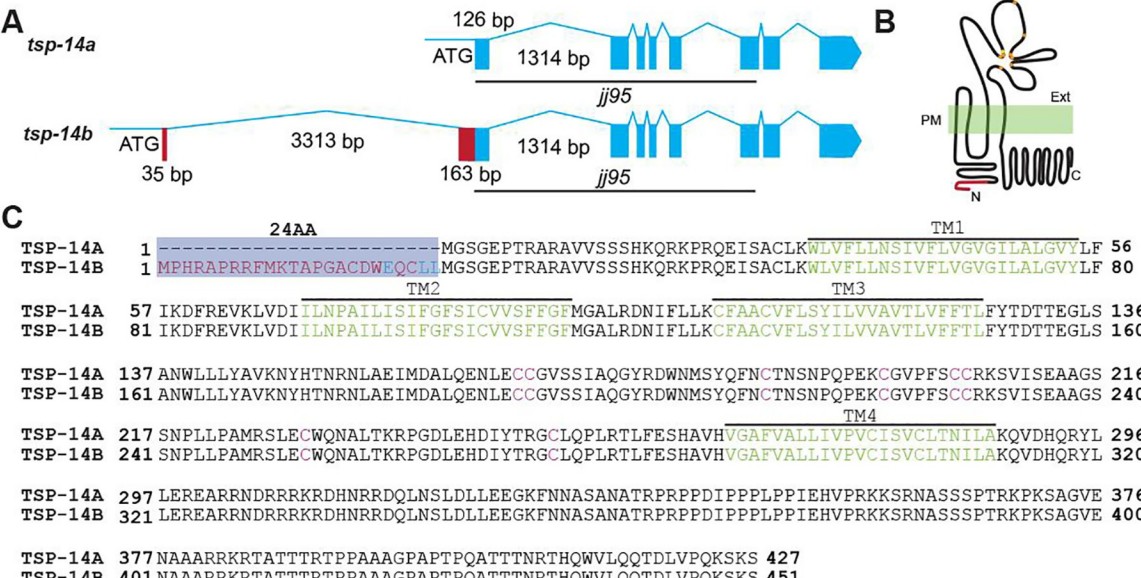

**Fig 1. The two TSP-14 isoforms: TSP-14A and TSP-14B. (A).** Schematic representation of the *tsp-14* locus, showing only the alternative and shared coding exons of *tsp-14a* and *tsp-14b*. Exons are represented by boxes, and introns by lines. ATG represents the start codon of each isoform. The first coding exon for *tsp-14b* is located 3511bp upstream of the first coding exon of *tsp-14a*. The black lines depict the extent of the *jj95* deletion, which is close to 2.9kb long. **(B).** A schematic of TSP-14A and TSP-14B proteins, showing the four transmembrane domains, and the various extracellular and intracellular regions. The only difference between TSP-14A and TSP-14B is at their N-termini, with the red line representing the extra amino acids present in TSP-14B. The CCG and CC motifs in the second extracellular loop are represented by solid circles, with orange solid circle representing cysteine, while the yellow solid circle representing glycine. PM, plasma membrane. Ext, extracellular. **(C).** Alignment of the amino acid sequences of TSP-14A and TSP-14B. TSP-14B has 24 additional amino acids at its N-terminus (red), which includes the ExxLL motif (blue). The four transmembrane domains are indicated in green and the cysteine residues in the extracellular domain are in red.

## TSP-14A and TSP-14B have distinct and shared functions during *C. elegans* development

To determine the functional significance of the two TSP-14 isoforms, we used CRISPR/Cas9 to generate isoform specific knock-outs, *tsp-14a(0)* and *tsp-14b(0)* (Figs 2A and S1A). Because the start codon of TSP-14A also encodes a methionine residue in TSP-14B, we mutated it from methionine (ATG) to isoleucine (ATA) to generate the *tsp-14a* null allele *jj304(tsp-14a(0))*. We reasoned that this methionine to isoleucine change can prevent the translation of TSP-14A without significant disruption of the TSP-14B coding sequence (Figs 2A and S1A). To generate the *tsp-14b* null allele *jj317(tsp-14b(0))*, we mutated the ATG start codon of TSP-14B and a downstream in-frame ATG codon to TTG and TTA respectively (Figs 2A and S1A) to prevent any in-frame TSP-14B-like product being produced. Because *tsp-14* null mutants have no phenotype on their own, but display multiple phenotypes in the *tsp-12(0)* background due to the functional redundancy shared by *tsp-14* and *tsp-12* [13,15], we tested the functionality of *tsp-14a(0)* and *tsp-14b(0)* by introducing each allele into the *tsp-12* null background. For this purpose, we used a known deletion allele of *tsp-12*, *ok239*, as well as a new deletion allele *jj300* that we generated via CRISPR (S1B Fig). Both alleles will be denoted as *tsp-12(0)*, unless specified. As shown in Fig 2, *tsp-12(0); tsp-14(0)* double null mutants are small (Sma), vulvaless (Vul) or with protruding vulvae (Pvl), and exhibit 100% maternal effect embryonic lethality (EMB) and a high penetrance (80%) of suppression of the *sma-9(0)* coelomocyte defects (Susm) ([16], Fig 2B, 2F–2H, 2P). Like *tsp-12(0); tsp-14(0)* double null mutants, *tsp-12(0); tsp-14a(0)* mutants are also Sma, Vul/Pvl, and 100% EMB (Fig 2B, 2I–2K). But *tsp-12(0); tsp-14a(0)* mutants only exhibit a 19.4% penetrance of the Susm phenotype (Fig 2P), a very slight increase of penetrance than that of *tsp-12(0)* single null mutants (16.5%). Conversely, *tsp-12(0); tsp-14b(0)* mutants have a normal body size, do not exhibit any vulva defects or embryonic lethality, yet exhibit a 37% penetrance of the Susm phenotype (Fig 2B, 2L–2N, 2P). These results suggest that TSP-14A and TSP-14B have distinct functions during *C. elegans* development: TSP-14A shares redundant functions with TSP-12 in regulating body size, embryonic and vulva development, while TSP-14B functions redundantly with TSP-12 in regulating postembryonic mesoderm development. Because *tsp-12(0); tsp-14(0)* double null mutants exhibit a 80% penetrance of the Susm phenotype, higher than that of either *tsp-12(0); tsp-14a(0)* or *tsp-12(0); tsp-14b(0)* mutants (Fig 2P), TSP-14A and TSP-14B must share redundant functions with each other, and with TSP-12, in regulating postembryonic mesoderm development.

## Single-copy transgene analysis supports distinct and overlapping functions of TSP-14A and TSP-14B

One caveat with the CRISPR-mediated knock-out experiments is that certain editing manipulations of one isoform could unavoidably affect the other one. To overcome this problem, we expressed TSP-14A or TSP-14B as single copy transgenes into the same neutral genomic environment [18], which allows for direct comparison of the functionality of the two isoforms. We chose the *ttTi4348* locus on Chromosome I as the insertion site and examined the function of each isoform on its own in the *tsp-12(0); tsp-14(0)* double null background. We used two different *tsp-14* promoter elements to drive the expression of cDNAs encoding either TSP-14A or TSP-14B: a 3.3kb promoter element, which is immediately upstream of the ATG of TSP-14A, and a 5.2kb promoter element, which includes another 1.9kb sequence upstream of the ATG of TSP-14B (S2 Fig, see also Material and Methods). Both promoter elements can drive GFP reporter expression in multiple cell types, including hypodermal cells (hyp 7 and seam cells), although we detected stronger and less mosaic reporter expression under the 5.2kb promoter (S2B and S2C Fig). Using both the 3.3kb and the 5.2kb promoter elements to drive the

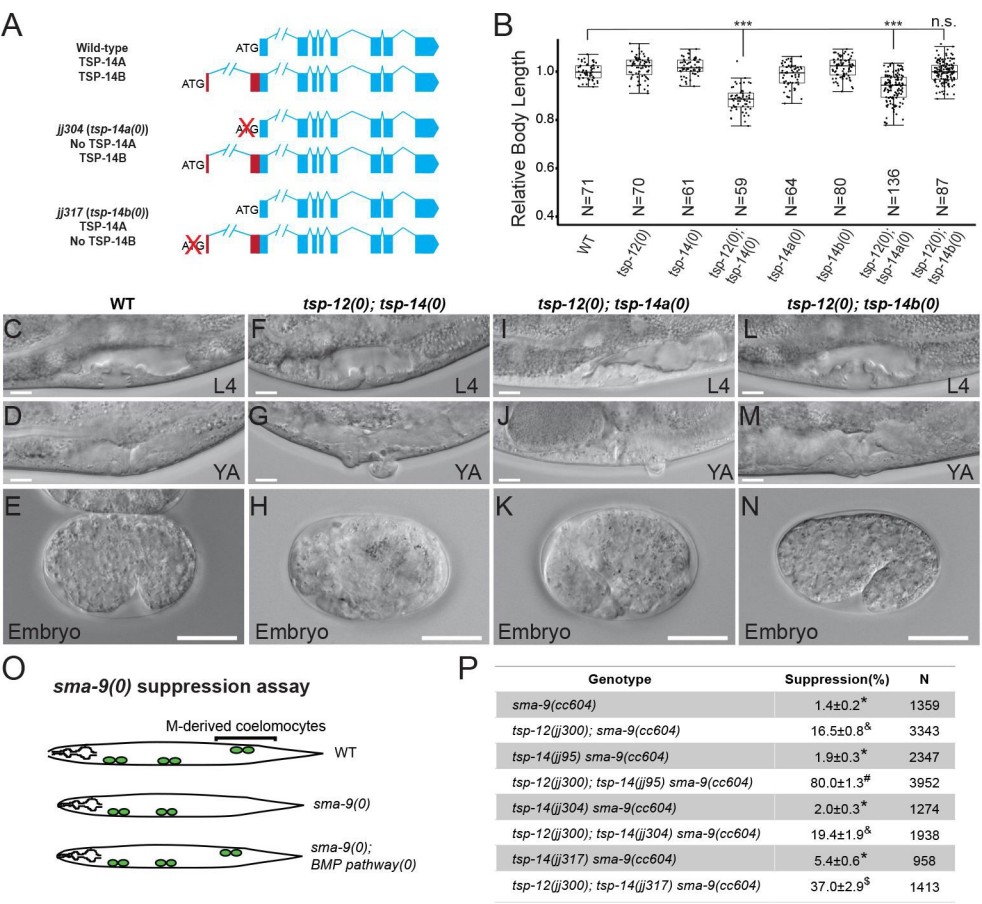

**Fig 2. Isoform-specific knockouts of TSP-14 cause distinct phenotypes. (A).** Schematics of the *tsp-14* locus, showing isoform-specific knockouts generated in this study. In *jj304*, the TSP-14A start codon ATG (methionine) is mutated to TTG (isoleucine), knocking out TSP-14A, while introducing a single, although conserved amino acid change in TSP-14B. The *tsp-14(jj304)* allele is also referred to as *tsp-14a(0)*. In *jj317*, the TSP-14B start codon ATG and a downstream, in-frame, ATG (see Fig 1C) are mutated to TTG and TTA respectively, knocking out TSP-14B without affecting TSP-14A. The *tsp-14(jj317)* allele is also referred to as *tsp-14b(0)*. **(B).** Relative body lengths of synchronized L4 worms of various genotypes. The mean body length of wild-type worms is normalized to 1.0. For each double mutant, data were pooled from two independent isolates. The total number of worms measured for each genotype is at least 60. Tukey's honestly significant difference (HSD) test following an ANOVA was used to test for differences between different genotypes. ***$P < 0.0001$. n.s., no significant difference. WT: wild-type. *tsp-12(0)*: *tsp-12(jj300)*. *tsp-14(0)*: *tsp-14(jj95)*. *tsp-14a(0)*: *tsp-14(jj304)*. *tsp-14b(0)*: *tsp-14(jj317)*. **(C-N).** Differential interference contrast (DIC) micrographs of wild-type (WT, C-E), *tsp-12(0); tsp-14(0)* (F-H), *tsp-12(0); tsp-14a(0)* (I-K), *tsp-12(0); tsp-14b(0)* (L-N) animals at the L4 (C, F, I, L) or young adult stage (D, G, J, M), or their embryos at mid-embryogenesis (E, H, K, N). L4, L4 larva. YA, young adult. Scale bars represent 20 μm. For each genotype, more than 100 animals or embryos were examined. **(O).** Diagrams depicting the *sma-9(0)* suppression (Susm) assay. Mutations in the BMP pathway can suppress the loss of M-lineage-derived coelomocytes (CCs) in *sma-9(0)* mutants. CCs are depicted as green circles. **P).** Table summarizing the results of the *sma-9(0)* suppression assay of various *tsp-14* alleles. Percentage of suppression was calculated by the number of worms with 1–2 M-derived CCs divided by the total number of worms scored. N represents the total number of worms counted. Data from two independent isolates were combined for each genotype. Groups marked with distinct symbols are significantly different from each other ($P<0.001$, in all cases when there is a significant difference), while groups with the same symbol are not. Tested using an ANOVA with a Tukey HSD (see Material and Methods).

expression of *tsp-14* as single copy transgenes, TSP-14A failed to rescue, while TSP-14B partially rescued, the Susm phenotype of *tsp-12(0); tsp-14(0)* double mutants (Fig 3A and 3B). This is consistent with the notion that TSP-14B plays a major role in postembryonic mesoderm development, but normal postembryonic development requires both TSP-14A and TSP-

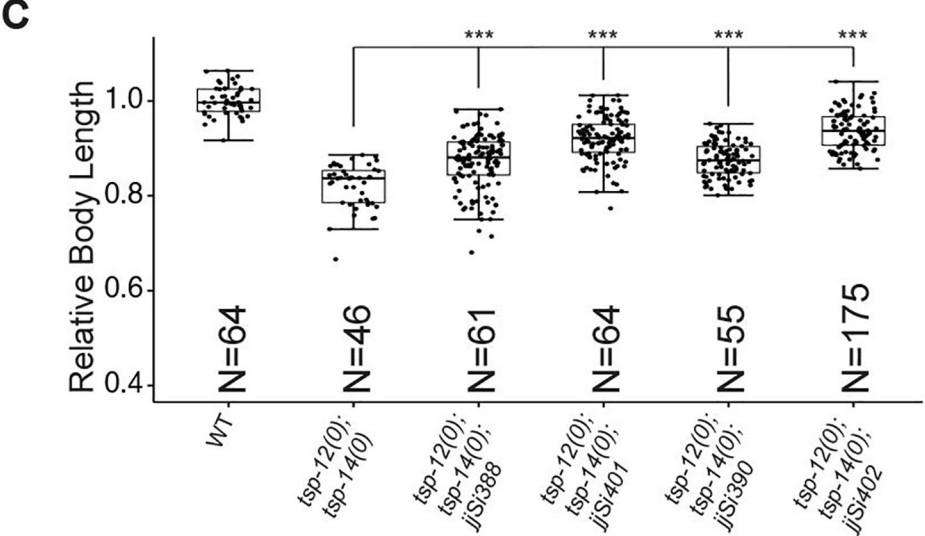

**Fig 3. TSP-14A and TSP-14B have distinct and shared functions. (A)** Schematics depicting the single copy transgenes expressing either *tsp-14a* or *tsp-14b* under the 3.3kb or 5.2kb *tsp-14* promoter at the *ttTi4348* locus on chromosome I. **(B)** Table summarizing the results of the *sma-9(0)* suppression assay when *tsp-14a* or *tsp-14b* was expressed as a single copy transgene. Percentage of suppression was calculated by the number of worms with 1–2 M-derived CCs divided by the total number of worms scored. Groups marked with distinct symbols are significantly different from each other (*P*<0.001, in all cases when there is a significant difference), while groups with the same symbol are not. Tested using an ANOVA with a Tukey HSD (see Material and Methods). **(C)** Relative body lengths of synchronized L4 worms of various genotypes at the Christmas tree stage. The mean body length of wild-type (WT) worms is normalized to 1.0. Numbers within each bar indicate the total number of worms measured. For each double mutant, data were pooled from two independent isolates. The total number of worms measured for each genotype is at least 60. Tukey's HSD test following an ANOVA was used to test for differences between different genotypes. ***$P < 0.0001$. WT: wild-type. *tsp-12(0)*: *tsp-12(jj300)*. *tsp-14(0)*: *tsp-14(jj95)*.

14B as well as TSP-12. Intriguingly, TSP-14A and TSP-14B each on its own can significantly rescue the body size defects of *tsp-12(0); tsp-14(0)* double mutants, with more efficient rescue observed when expression was driven by the longer 5.2kb *tsp-14* promoter as compared with the 3.3kb promoter (Fig 3C). Taken together, the above results support the notion that TSP-14A and TSP-14B exert distinct and overlapping functions in regulating body size and mesoderm development.

## Endogenous TSP-14A and TSP-14B exhibit distinct expression and localization patterns

To understand the basis underlying the distinct and shared functions of TSP-14A and TSP-14B, we used CRISPR/Cas9-mediated homologous recombination to tag each of the isoforms and examined their expression and subcellular localization patterns. We used two different approaches to tag each of the isoforms.

First, we inserted GFP and three copies of the FLAG tag (3xFLAG) at their respective N-termini after the ATG start codons. This led to the generation of N-terminally tagged TSP-14B (*jj192*, TSP-14B(NT) from here on) and TSP-14A (*jj183*, TSP-14A(NT) from here on, see below) (Fig 4A, Material and Methods). We verified the successful tagging of each isoform by western blot analysis, which also indicated that TSP-14A is expressed at a much higher level compared to TSP-14B (Fig 4D). Furthermore, although both TSP-14A and TSP-14B are expected to be tagged in *jj183* animals, we did not detect any tagged TSP-14B product on western blots. Because the first coding exon of TSP-14A is part of the second coding exon of TSP-14B (Fig 4A), it is possible that the GFP tag in *jj183* animals affects the proper expression or splicing of TSP-14B.

We then examined the phenotypes of TSP-14A(NT) (*jj183*) and TSP-14B(NT) (*jj192*) animals. As expected, both TSP-14A(NT) and TSP-14B(NT) animals are healthy and fertile, although TSP-14A(NT) worms are slightly longer than wild-type worms at the same developmental stage (Fig 4B). We then introduced TSP-14A(NT) and TSP-14B(NT) into the *tsp-12(0)* null background, and examined their phenotypes. *tsp-12(0); TSP-14A(NT)* and *tsp-12(0); TSP-14B(NT)* animals are both viable and fertile, without any embryonic lethality (Figs 4B–4C and S3), suggesting that neither tag significantly disrupts TSP-14 function. *tsp-12(0); TSP-14A(NT)* animals have a normal body size (Figs 4B and S3A), but exhibit a slight increase in penetrance of the Susm phenotype compared to *tsp-12(0)* single mutants (Figs 4C and S3B). We suspect that the slight increase in the penetrance of the Susm phenotype in *tsp-12(0); TSP-14A(NT)* animals is due to the reduced or lack of expression of TSP-14B. Surprisingly, *tsp-12(0); TSP-14B(NT)* mutants have a significantly lower penetrance of the Susm phenotype compared to that of *tsp-12(0)* single mutants (Figs 4C and S3B). This improved Susm phenotype may be due to the higher expression level of tagged TSP-14B(NT) compared to the endogenous level of TSP-14B (Fig 4D, see below).

We have previously tagged TSP-14 endogenously at its C-terminus with GFP::3xFLAG (*jj219*, TSP-14AB(CT) from here on, Fig 3A) and showed that *jj219*, which has both TSP-14A and TSP-14B tagged, is fully functional [16]. Therefore, as an alternative approach to specifically tag TSP-14A or TSP-14B, we introduced the same GFP::3xFLAG tag as in *jj219* into either *jj304(tsp-14a (0))* or *jj317(tsp-14b(0))*, and generated *jj304 jj319*, which has C-terminally tagged TSP-14B without any TSP-14A (TSP-14B(CT) from here on), and *jj317 jj377*, which has C-terminally tagged TSP-14A without any TSP-14B (Fig 4A, TSP-14A(CT) from here on). Both *jj304 jj319* and *jj317 jj377* animals behaved similarly as their respective untagged counterparts, *jj304(tsp-14a(0))* and *jj317(tsp-14b(0))*. Western blots confirmed the specific tagging of TSP-14 isoforms in *jj304 jj319* and *jj317 jj377* animals (Fig 4D). Notably, animals carrying TSP-14B(NT) appear to have an elevated amount of TSP-14B protein than animals carrying either TSP-14B(CT) or animals carrying TSP-14AB(CT) (Fig 4D). The underlying basis is currently unknown.

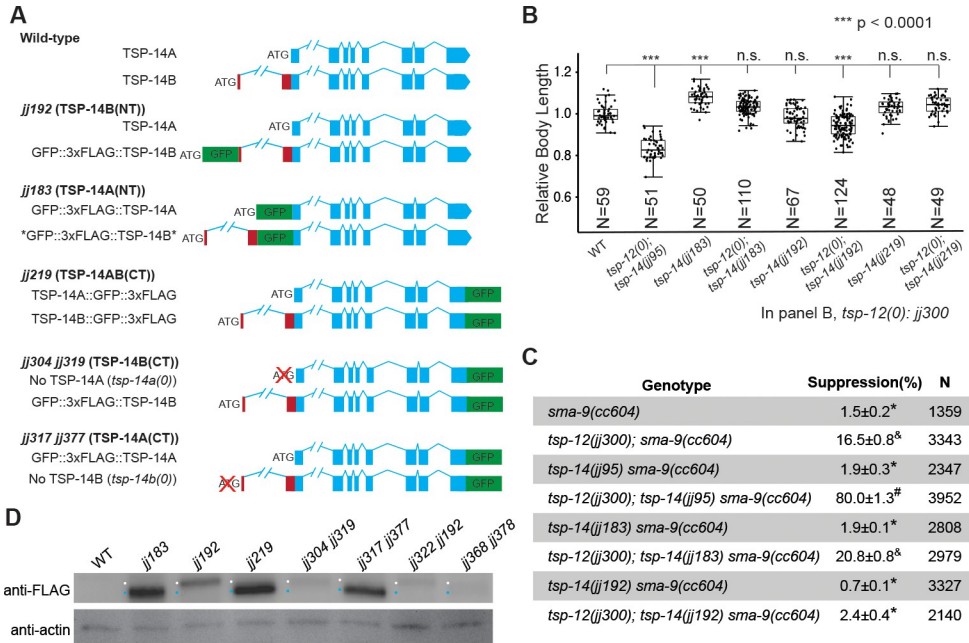

| Genotype | Suppression(%) | N |
|---|---|---|
| sma-9(cc604) | 1.5±0.2* | 1359 |
| tsp-12(jj300); sma-9(cc604) | 16.5±0.8& | 3343 |
| tsp-14(jj95) sma-9(cc604) | 1.9±0.3* | 2347 |
| tsp-12(jj300); tsp-14(jj95) sma-9(cc604) | 80.0±1.3# | 3952 |
| tsp-14(jj183) sma-9(cc604) | 1.9±0.1* | 2808 |
| tsp-12(jj300); tsp-14(jj183) sma-9(cc604) | 20.8±0.8& | 2979 |
| tsp-14(jj192) sma-9(cc604) | 0.7±0.1* | 3327 |
| tsp-12(jj300); tsp-14(jj192) sma-9(cc604) | 2.4±0.4* | 2140 |

**Fig 4. Endogenous TSP-14A ad TSP-14B can be respectively and functionally tagged. (A).** Schematic representations of wild-type and GFP-tagged TSP-14 isoforms. In *jj192* (denoted as TSP-14B(NT)) animals, TSP-14B is specifically tagged at its N-terminus with GFP::3xFLAG. In *jj183* (denoted as TSP-14A(NT)) animals, TSP-14A specifically tagged at its N-terminus with GFP::3xFLAG. * If spliced properly, the tag should also be inserted into TSP-14B after the first 24aa in *jj183* animals. However, no tagged TSP-14B was detectable on western blots (**D**), suggesting that the expression or splicing of TSP-14B is compromised in *jj183* animals. Both TSP-14A and TSP-14B are tagged at their C-termini with GFP::3xFLAG in *jj219* (denoted as TSP-14AB(CT) animals. The same GFP::3xFLAG tag as in *jj219* is also in *jj304 jj319* (denoted as TSP-14B(CT) and *jj317 jj377* (denoted as TSP-14A(CT) animals when one of the respective *tsp-14* isoforms is knocked out. (**B**). Relative body lengths of synchronized L4 worms of various genotypes. The mean body length of wild-type worms is normalized to 1.0. For each double mutant, data were pooled from two independent isolates. The total number of worms measured for each genotype is at least 60. Tukey's HSD test following an ANOVA was used to test for differences between different genotypes. ***$P < 0.0001$. n.s., no significant difference. As shown, *tsp-12(0); tsp-14(jj192)* worms are slightly smaller than wild-type (WT) worms, but not as small as *tsp-12(0); tsp-14(0)* worms. (**C**). Tables summarizing the results of the *sma-9(0)* suppression assay of various *tsp-14* alleles, in combination with the *tsp-12(jj300)* null allele. Percentage of suppression was calculated by the number of worms with 1–2 M-derived CCs divided by the total number of worms scored. N represents the total number of worms counted. Data from two independent isolates were combined for each genotype. Groups marked with distinct symbols are significantly different from each other ($P<0.001$, in all cases when there is a significant difference), while groups with the same symbol are not. Tested using an ANOVA with a Tukey HSD (see Material and Methods). (**D**). Western blot showing the various tagged forms of TSP-14A (blue dot) or TSP-14B (white dot) detected by anti-FLAG antibodies. Anti-actin antibodies were used as a loading control. For each lane, 100 synchronized L4 animals were used to prepare the samples. The genotypes of *jj322 jj192* and *jj368 jj378* are described in Fig 7.

Using the various strains generated above. we conducted high resolution imaging using the Aryscan imaging system (Materials and Methods). We found that TSP-14A and TSP-14B exhibit distinct expression and localization patterns. First, only TSP-14A, but not TSP-14B, is detectable in the germline and sperm cells, as well as at the tip of the anterior sensory cilia (Fig 5A and 5B), while TSP-14B is detectable in the pharynx (Fig 5F). Both TSP-14A and TSP-14B are found in hypodermal cells (Fig 5C, 5G, 5K, 5P, 5S, 5V) and the developing vulva (Fig 5E, 5I, 5M, 5O, 5R, 5U). However, the two isoforms exhibit different subcellular localization patterns. In hypodermal cells, TSP-14A is primarily localized in intracellular vesicles, while TSP-14B is mainly localized on the cell surface (compare Fig 5C vs. 5G, and Fig 5S vs. 5V, S1–S3 Movies). In the developing vulva, TSP-14A is localized on the apical side, while TSP-14B is localized on the basolateral membrane (compare Fig 5E vs. 5I, and Fig 5R vs. 5U,

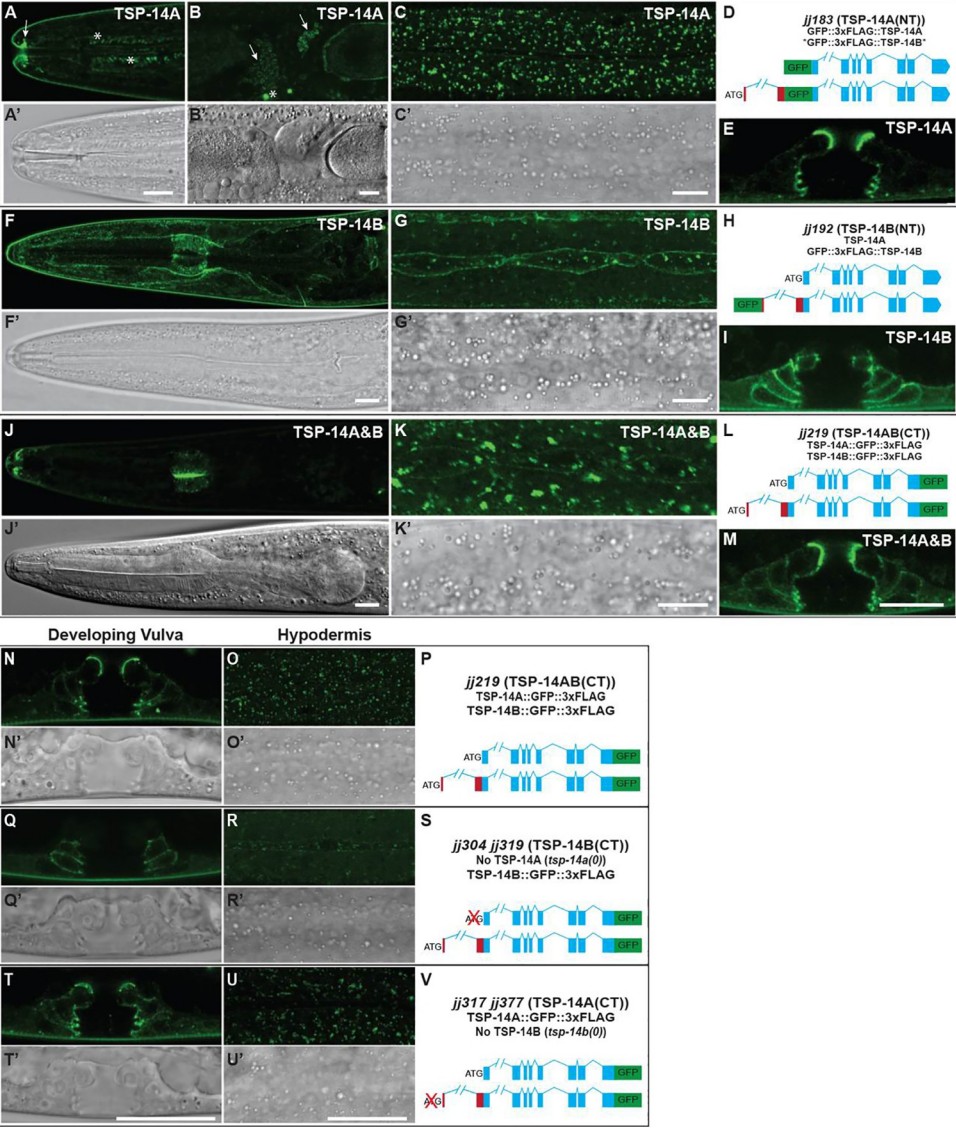

**Fig 5. TSP-14A and TSP-14B exhibit distinct localization patterns. (A-M)** Airyscan confocal images showing the localization of TSP-14A (A-C' and E), TSP-14B (F-G' and I), and both TSP-14A and TSP-14B (J-K' and M) in the head region (GFP: A, F, J, and DIC: A', F', J'), hypodermal cells (GFP: C, G, K, and DIC: C', G', K'), and the developing vulva (E, I, and M). TSP-14A(NT) is detectable in the tips of anterior sensory cilia (arrow in A), and in sperm cells (arrows in B). * in A and B marks autofluorescence signals. Scale bar in A-M, 10 μm. Panels D, E and L are schematic representations of the different knock-ins. (**N-V**) Airyscan confocal GFP (N, O, Q, R, T, U) and corresponding DIC (N', O', Q', R', T', U') images of the developing vulva (N, Q, T) and hypodermal cells (O, R, U) from worms expressing different tagged versions of TSP-14. Panels P, S and V are schematic representations of the different knock-ins. Scale bars in O-W, 20 μm.

for worms at the L4 Christmas-tree stage). Similar localization patterns hold true in the pharynx (Fig 5A, 5F and 5J). Consistent with the higher levels of TSP-14A protein that we detected on western blots (Fig 4E), the TSP-14A signals are significantly brighter than TSP-14B under the microscope (see Fig 5J, 5K, 5M and 5O when both TSP-14A and TSP-14B are present). Thus, TSP-14A and TSP-14B exhibit distinct expression and localization patterns.

## TSP-14A is localized to early and late endosomes

We have previously shown that TSP-14 is localized to early, late, and recycling endosomes, that it co-localizes with TSP-12 and shares functional redundancy with TSP-12 in promoting BMP signaling [16]. Because TSP-14A is localized to intracellular vesicles, we further examined its localization relative to various endosomal markers and to TSP-12. As shown in Fig 6,

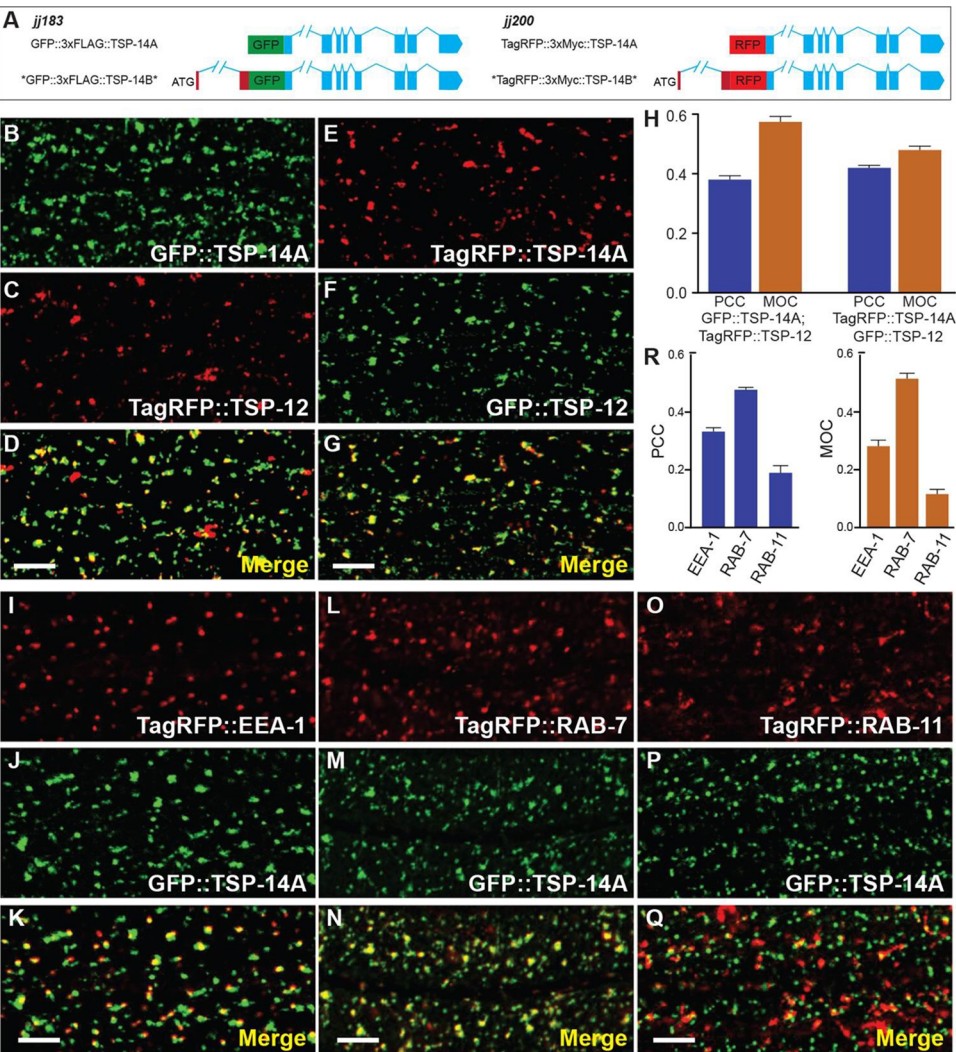

**Fig 6. TSP-14A is co-localized with TSP-12 and is localized to endosomes.** (**A**) Schematics of GFP- or TagRFP-tagged TSP-14A. (**B-G**) Airyscan confocal images of late L4 hypodermal cells expressing both tagged TSP-14A (B, GFP-tagged TSP-14A and E, TagRFP-tagged TSP-14A) and tagged TSP-12 (C, TagRFP-tagged TSP-12 and F, GFP-tagged TSP-12). The corresponding merged images are shown in D and G, respectively. (**H**) The colocalization between TSP-14A and TSP-12 was quantified using two different methods. PCC, Pearson's correlation coefficient; MOC, Mander's overlap coefficient. PCC indicates the correlation of GFP::TSP-14A with TagRFP::TSP-12 or TagRFP:: TSP-14A with GFP::TSP-12. MOC indicates the fraction of TagRFP::TSP-12 that overlaps with GFP::TSP-14A (the second bar), or the fraction of GFP::TSP-12 that overlaps with TagRFP::TSP-14A (the forth bar). (**I-Q**) Airyscan confocal images of L4 stage hypodermal cells expressing N-terminally GFP-tagged TSP-14A (J, M, P) and different endosomal markers, TagRFP::EEA-1 (I, early endosomes), TagRFP::RAB-7 (L, late endosomes and lysosomes), and TagRFP::RAB-11 (O, recycling endosomes). K, N and Q are the corresponding merged images. (**R**) Bar graphics showing the correlation of TagRFP (EEA-1, RAB-7 or RAB-11) with GFP::TSP-14A (PCC), and the fraction of TSP-14A::GFP that overlaps with TagRFP (MOC). Error bars in H and R represent SEM. Statistical analysis was done using a 2-tailed $t$ test with 95% CIs. Scale bar, 5 μm.

TSP-14A co-localizes with TSP-12 (Fig 6B–6H). Furthermore, TSP-14A co-localizes with both the early endosome marker RAB-5 and the late endosome/early lysosome marker RAB-7, but shows little co-localization with the recycling endosome marker RAB-11 (Fig 6I–6R). Given that we previously detected recycling endosome localization of TSP-14 in TSP-14AB(CT) (*jj219*) animals [16], our findings suggest that the very faint vesicular signal from TSP-14B may come from recycling endosomes. However, due to the very weak signal of TSP-14B and the limited number of TSP-14B-positive vesicles, we have not been able to address this question using the reagents that we generated. Nevertheless, our findings suggest that TSP-14A is localized to different endosomal vesicles.

## TSP-14B is sufficient on its own to localize to the basolateral membrane of polarized cells

Our finding that TSP-14B(CT) in *tsp-14a* null (*jj304 jj319*) animals is localized to the basolateral surface of vulval and hypodermal cells (Fig 5R–5T) suggests that the basolateral localization of TSP-14B does not require the presence of TSP-14A. To directly test this hypothesis, we forced the expression of C-terminally-tagged TSP-14B as a single copy transgene (*jjSi395*) in *tsp-14(0)* null background using a heterologous *snx-1* promoter. As a control, we also forced the expression of C-terminally-tagged TSP-14A (*jjSi393*) using the same approach. In the *tsp-14(0)* null background, TSP-14B (*jjSi395)* is localized to the basolateral membrane of pharyngeal and intestinal cells, while the TSP-14A (*jjSi393*) signal is only weakly detectable in intracellular vesicles in the intestine (Fig 7A and 7B). *jjSi393* and *jjSi395* animals did not show strong GFP signals in hypodermal cells or in the vulva, presumably due to the *tbb-2* 3'UTR used in our expression constructs (Fig 7), which is optimal for germline transgene expression [19]. Nevertheless, our data showed that in the absence of TSP-14A, TSP-14B alone is sufficient to localize to the basolateral membrane of polarized cells. This result suggests that the basolateral membrane targeting sequence resides in the coding region of TSP-14B.

## TSP-14B contains a basolateral membrane targeting signal in its first 24 amino acids

Since the only difference between TSP-14A and TSP-14B is the N-terminal 24 amino acids present in TSP-14B, we reasoned that these 24 amino acids may contain signal(s) that targets TSP-14B to the basolateral membrane. Indeed, we found an EQCLL motif (Fig 8A) within these 24 amino acids in TSP-14B, similar to the well characterized di-leucine basolateral targeting sequence [DE]xxxL[LI] [20,21]. Using the CRISPR/Cas9 system, we mutated the EQCLL sequence into AQCAA in the TSP-14B(NT) background (Fig 8B) and obtained *tsp-14(jj322 jj192)* (Figs 8E and S1C). We found that the GFP-tagged AQCAA mutant TSP-14B protein in *jj322 jj192* worms becomes localized to the apical side of the developing vulva (compare Fig 8C vs 8F) and in intracellular vesicles of hypodermal cells (Fig 8D vs. 8G), just like TSP-14A. These results demonstrate that TSP-14B's localization to the basolateral membrane depends on the EQCLL motif.

To examine the functional consequences of having TSP-14B mis-localized, we introduced the AQCAA mutation into wild-type worms via CRISPR and obtained *tsp-14(jj368)* (Figs 8A and 1C). We then introduced *jj368* into the *tsp-12(0)* background and examined the phenotypes of *tsp-12(0); tsp-14(jj368)* double mutants. We found that *tsp-12(0); tsp-14(jj368)* double mutants [*tsp-12* null mutants with mis-localized TSP-14B] exhibit more severe EMB, body size and and more penetrant Susm phenotypes compared to *tsp-12(0); tsp-14(jj317)* double mutants [*tsp-12* null mutants without any TSP-14B] (compare Fig 8H vs. Fig 2B, and compare Fig 8I vs. Fig 2P). Moreover, *tsp-12(0); tsp-14(jj368)* double mutants exhibit a significantly

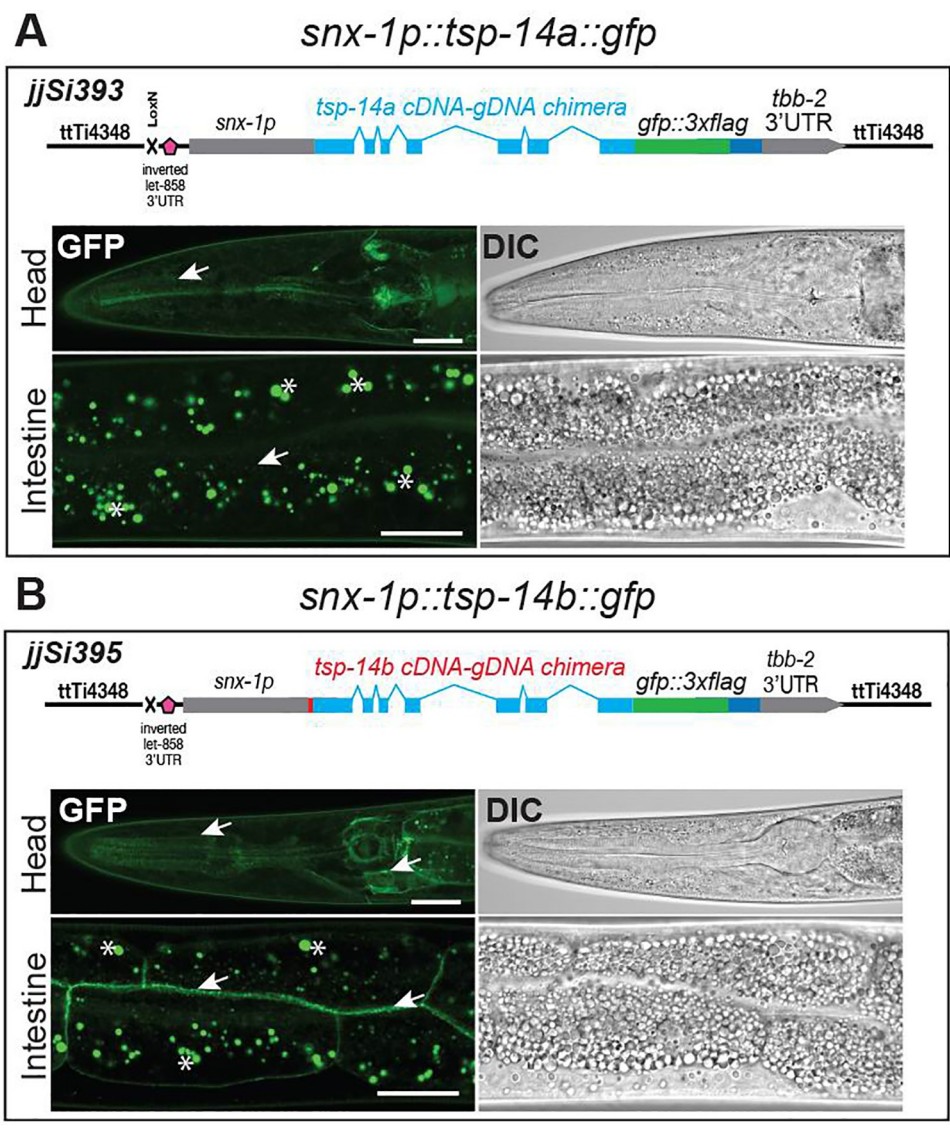

**Fig 7. TSP-14B is sufficient to localize to the basolateral membrane upon forced expression.** Airyscan confocal images of L4 worms expressing either TSP-14A (**A**) or TSP-14B (**B**) in the *tsp-14(0)* background under the *snx-1* promoter as single copy transgenes in the *ttTi4348* locus on chromosome I. Images shown are of the head region and intestine cells. Arrows point to TSP-14A or TSP-14B signals. Asterisks mark intestinal autofluorescence. TSP-14B::GFP is concentrated on the membrane in the head region and in the basolateral membrane of the intestinal cells, while the TSP-14A::GFP signal is not detectable on cell membranes. Scale bars, 20 μm.

reduced brood size ([Fig 8J]) and severe embryonic lethality (77% EMB), phenotypes not displayed by *tsp-12(0); tsp-14(jj317)* double mutants. We reasoned that these phenotypes seen in the *tsp-12(0); tsp-14(jj368)* double mutants cannot simply be due to the mis-localization of TSP-14B. Because the EQCLL motif is located right upstream of the ATG codon of TSP-14A ([Fig 8A]), mutating this motif may have affected a cis-regulatory element(s) important for the proper expression of TSP-14A. To test this hypothesis, we tagged both TSP-14A and TSP-14B in the *jj368* background at their C-terminal ends using CRISPR, and generated *tsp-14(jj368 jj378*, [Fig 8K]). As suspected, *tsp-14(jj368 jj378)* worms (expressing AQCAA-TSP-14AB(CT)) have very little TSP-14A or TSP-14B protein that is detectable on western blots ([Fig 4D]) and

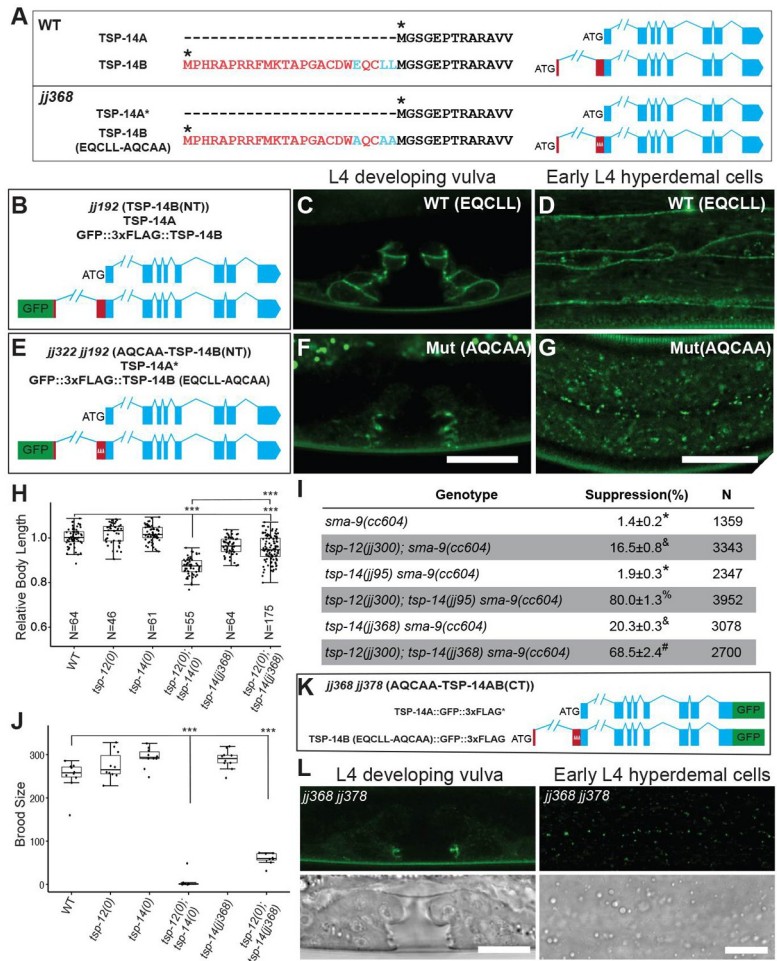

**Fig 8. The 24aa unique to TSP-14B contains a basolateral membrane targeting sequence. (A, B, E)** Schematics of the wild-type (WT) and *jj368* mutant *tsp-14* loci, either untagged (A), or N-terminally GFP-tagged (B, E), showing the EQCLL sequence in WT TSP-14B (or in *jj192*, B) mutated to AQCAA in *jj368* (A) or *jj322 jj192* (E). **(C-D, F-G)** Airyscan confocal images of L4 larvae showing that wild-type (WT) TSP-14B is localized to the basolateral membrane in the developing vulva (C) and in hypodermal cells (D), while mutated TSP-14B carrying the AQCAA mutation is localized to the apical side of the developing vulva (F) and in intracellular vesicles in hypodermal cells (G). Scale bars, 10 μm. (**H**) Relative body lengths of synchronized L4 worms at the Christmas tree stage. The mean body length of wild-type worms is normalized to 1.0. For each double mutant, data were pooled from two independent isolates. The total number of worms measured for each genotype is at least 60. Tukey's HSD following an ANOVA was used to test for differences between different genotypes. ***$P < 0.0001$. WT: wild-type. *tsp-12(0)*: *tsp-12(jj300)*. *tsp-14(0)*: *tsp-14 (jj95)*. As shown, *tsp-12(0); tsp-14(jj368)* worms are slightly smaller than WT worms, but not as small as *tsp-12(0); tsp-14(0)* worms. (**I**) Table summarizing the results of the *sma-9(0)* suppression assay. Percentage of suppression was calculated by the number of worms with 1–2 M-derived CCs divided by the total number of worms scored. N represents the total number of worms counted. Data from two independent isolates were combined for each genotype. Groups marked with distinct symbols are significantly different from each other ($P<0.001$, in all cases when there is a significant difference), while groups with the same symbol are not. Tested using an ANOVA with a Tukey HSD (see Material and Methods). (**J**) Graph summarizing the brood sizes of different strains. *tsp-12(0)*, *tsp-14(0)* and *tsp-14 (jj368)* single mutants have the same brood size as wild-type (WT) worms, while *tsp-12(0); tsp-14(0)* and *tsp-12(0); tsp-14(jj368)* double mutants have significant reduced brood sizes compared to WT. *tsp-12(0)*: *tsp-12(jj300)*. *tsp-14(0)*: *tsp-14(jj95)*. Tukey's HSD following an ANOVA was used to test the differences between different genotypes. ***$P < 0.0001$. (**K**) Schematic representations of the N-terminally GFP-tagged TSP-14A and N-terminally GFP-tagged TSP-14A with EQCLL mutated to AQCAA (*jj368 jj378*). (L) Airyscan confocal images of L4 larvae showing the developing vulva and hypodermal cells. The GFP signal in *jj368 jj378* worms is very faint and not localized to the basolateral cell membrane. Scale bars, 10 μm.

via imaging (Fig 8L). Any detectable TSP-14 protein in *tsp-14(jj368 jj378)* animals appears to be apical and intracellularly localized (Fig 8L). We noticed that *tsp-14(jj368)* single mutants exhibit a 20% Susm phenotype, while *tsp-14(0)* null mutants do not show any Susm phenotype (Fig 8I), suggesting that mis-targeting TSP-14B to the apical side may have a dominant negative effect on TSP-14 and TSP-12 function.

## Discussion

Despite extensive biochemical studies of the tetraspanin family of proteins, there are significant gaps in our understanding of how different tetraspanin proteins function in vivo in different cellular and developmental contexts. A major challenge for dissecting the in vivo functions of multi-member families of proteins, such as the tetraspanins, in vertebrates is the functional redundancy shared by different members of the same family or subfamily, or by different isoforms of the same protein (for example, [22,23]). In this study, we used CRISPR-mediated knock-in and knock-out technology and showed that two isoforms of a single *C. elegans* TspanC8 tetraspanin, TSP-14A and TSP-14B, exhibit distinct subcellular localization patterns, and share overlapping as well as unique expression patterns and functions. Our work highlights the diverse and intricate functions of TspanC8 tetraspanins in a living organism. It also adds another example to the existing literature that shows protein isoforms are another way for *C. elegans* to increase the diversity and versatility of its proteome.

Our isoform specific knock-in experiments showed that TSP-14B is localized to the basolateral membrane, while TSP-14A is localized to the early and recycling endosomes on the apical side of polarized epithelial cells. We further identified a basolateral membrane targeting sequence (EQCLL) within the N-terminal 24 amino acids unique to TSP-14B, and showed that this ExxLL motif is critical for the basolateral membrane localization of TSP-14B. The ExxLL di-leucine motif in TSP-14B differs slightly from the canonical [DE]xxxL[LI] di-leucine motif, which has been previously found to serve as a basolateral membrane targeting sequence (for reviews, see [20,21]). Basolateral sorting of proteins with the canonical [DE]xxxL[LI] di-leucine motif is primarily mediated by the AP-2 clathrin adaptor, although it can also be mediated by the AP-1 adaptor (for reviews, see [20,21]). Both AP-1 and AP-2 adaptor complexes exist in *C. elegans* [24–27]. Future work will determine whether the AP-1 or the AP-2 adaptor is involved in sorting TSP-14B to the basolateral membrane.

To date, most TspanC8 tetraspanins have been reported to be either cell surface or intracellularly localized in cultured cells, most of which are non-polarized [5,8,12,28]. Our findings suggest that some TspanC8 proteins may also exhibit distinct localization patterns in polarized cells. Indeed, a recent report showed apical junction localization of the TspanC8 member Tspan33 in polarized mouse cortical collecting duct (mCCD) cells [9]. In the same study, the authors reported that another TspanC8 protein, Tspan15, is not on the apical plane, but is instead localized along lateral contacts of polarized mCCD cells [9]. Whether other TspanC8 tetraspanins or their specific isoforms also exhibit distinct subcellular localization patterns in polarized epithelial cells remains to be determined.

How the apically localized TSP-14A and basolateral membrane-localized TSP-14B exert their distinct, as well as overlapping, functions in vivo is currently not well understood. Nevertheless, our studies provide clues for the cellular basis underlying the functional redundancy shared by TSP-12 and TSP-14. TSP-14 is known to function redundantly with its paralog TSP-12 to promote both Notch signaling and BMP signaling [13,15,16]. Specifically, *tsp-12(0); tsp-14(0)* double null mutants exhibit maternal-effect embryonic lethality (Emb) and are egg-laying defective (Egl) due to defects in vulva development, two processes known to be regulated by Notch signaling [13]. *tsp-12(0); tsp-14(0)* double null mutants are also small and exhibit a

suppression of the *sma-9(0)* postembryonic mesoderm defect (Susm), processes regulated by BMP signaling [17,29]. Like TSP-14, TSP-12 is localized to various endosomes, as well as on the basolateral membrane of polarized vulval and intestinal epithelial cells [16]. Animals lacking TSP-12 and TSP-14A share the same penetrance (100%) of Emb and Egl phenotypes as *tsp-12(0); tsp-14(0)* double null animals, suggesting that endosome-localized TSP-12 and TSP-14A are critical for most, if not all, of the biological processes regulated by Notch signaling. In contrast, BMP-regulated body size and postembryonic mesoderm patterning appear to require both TSP-14A and TSP-14B to function together with TSP-12. Yet the contributions of TSP-14A and TSP-14B towards these two BMP-regulated processes are not equal. Three lines of evidence suggest that TSP-14B, but not TSP-14A, is the major player that functions redundantly with TSP-12 in patterning the postembryonic mesoderm. First, animals lacking TSP-12 and TSP-14B (but not TSP-14A) exhibit a higher penetrance of the Susm phenotype than that of *tsp-12(0)* single mutants (Fig 2). Second, when introduced as a single copy transgene, TSP-14B, but not TSP-14A, could partially rescue the Susm phenotype of *tsp-12(0); tsp-14(0)* double null mutants (Fig 3). Third, a moderate increase in the level of TSP-14B in TSP-14B(NT) was sufficient to partially compensate for the lack of TSP-12 in *tsp-12(0)* null mutants in patterning the postembryonic mesoderm (Figs 4 and S3). Consistently, when *tsp-14b* was overexpressed as a transgene in the *tsp-12(0); tsp-14(0)* double null background, the penetrance of the Susm phenotype is also lower than that seen in *tsp-12(0)* single mutants (S2 Fig). Despite this shared redundancy between TSP-12 and TSP-14B in mesoderm patterning, TSP-14A also plays a role in this process, as the penetrance of the Susm phenotype in *tsp-12(0); tsp-14(0)* double null mutants is significantly higher than that of *tsp-12(0); tsp-14b(0)* animals (Fig 2).

The roles of TSP-14A and TSP-14B in body size regulation appear more complicated. On the one hand, knockout experiments suggest that TSP-14A plays a major role in regulating body size, as *tsp-12(0); tsp-14a(0)* double mutants have a smaller body size, although they are not as small as the *tsp-12(0); tsp-14(0)* double null mutants (Fig 2). On the other hand, forced expression of either TSP-14A or TSP-14B can partially rescue the small body size phenotype of *tsp-12(0); tsp-14(0)* double null mutants (Fig 3). Furthermore, increased amount of TSP-14B in *tsp-12(0); TSP-14B(NT)* mutants [with endogenous TSP-14A but without TSP-12] led to both a slight decrease in body size and a less penetrant Susm phenotype (Figs 4B and 3A), suggesting possible antagonistic roles of TSP-14B in the regulation of body size vs. mesoderm development. These functional differences of TSP-14A and TSP-14B may be due to a combination of their unique expression and localization patterns, and ultimately, to the distinct molecular interactions at their specific cellular and subcellular environment.

Tetraspanins can have homotypic and heterotypic interactions to organize membranes into tetraspanin-enriched microdomains, and regulate the trafficking or clustering of different membrane or membrane-associated proteins [2,30–32]. For example, TspanC8 tetraspanins are known to bind and directly regulate the maturation and cell surface localization of ADAM10 [5–7], which is a metalloprotease that cleaves the Notch receptor in *C. elegans*, *Drosophila*, and mammals. We have previously shown that the *C. elegans* ADAM10, called SUP-17, is also involved in regulating BMP signaling, and that the neogenin homolog UNC-40 may be one of the SUP-17/ADAM10 substrates in regulating BMP signaling [15]. We further showed that TSP-12, but not TSP-14, is required for the cell surface localization of SUP-17/ADAM10 in the early embryo [15]. In addition, TSP-12 and TSP-14 function redundantly to promote the trafficking and cell surface localization of the BMP type II receptor DAF-4 in the developing larvae [16]. It will be important to determine the specific contributions that TSP-14A and TSP-14B each makes towards these TSP-14-mediated functions, and how each of the isoforms works at the molecular level. Finally, both TSP-12 and TSP-14 are also expressed in tissues beyond the sites where BMP signaling and Notch signaling act. They could in principle regulate the trafficking and/or localization of other

proteins in addition to SUP-17/ADAM10 and DAF-4/BMPRII. The identification of additional proteins trafficked by TSP-12 and TSP-14 (either TSP-14A or TSP-14B or both), and subsequent determination on how TSP-12 and TSP-14 regulate their trafficking/localization will elucidate how these two tetraspanin proteins function during the development of a multicellular living organism like *C. elegans*. The information obtained could shed light on the myriad of complex functions of the various TspanC8 proteins in mammals, and ultimately aid in the development of therapeutic strategies for the treatment of various diseases caused by abnormal expression or function of certain tetraspanins [33–35].

## Materials and methods

### *C. elegans* strains and molecular reagents

*C. elegans* strains used and generated in this study are listed in S1 Table. All *C. elegans* strains used in this study are derived from the Bristol N2 strain, maintained at 20°C (unless otherwise noted). Oligonucleotides and plasmids used and generated in this study are listed in S2 Table.

### Body size measurement and *sma-9(0)* suppression assays

Body size measurement and the *sma-9(0)* suppression assays were performed following the methods described in [36]. Hermaphrodite worms at the L4 Christmas-tree stage (based on the developing vulva) were used for body size measurement. Images of worms were taken using a Leica DMR2 compound microscope equipped with a Hamamatsu Orca-03G camera using the iVision software (BioVision Technologies). The length of each worm was then measured using segmented lines in the open-source software Fiji. Body length of at least 30 worms per genotype were measured. For the *sma-9(0)* coelomocyte suppression (Susm) assays, strains expressing the *CC::gfp* marker *arIs37(secreted CC::gfp)* or *ccIs4438(intrinsic CC::gfp)* in specific mutant backgrounds were generated (S1 Table). Hermaphrodite worms with GFP-labeled coelomocytes (CCs) were scored at the young adult stage under a Nikon SMZ1500 stereo zoom microscope equipped with a Sola light engine (Lumencor). For each genotype, two independent isolates were used and worms from three to five plates per isolate were scored. For each plate, the number of worms with 5–6 CCs was divided by the total number of worms examined to determine the penetrance of the Susm phenotype. For statistical analysis, an ANOVA and Tukey's honestly significant difference (HSD) test were performed to test differences in body size and Susm penetrance between different genotypes using R.

### CRISPR/Cas9 experiments

For all CRISPR/Cas9 experiments, Cas9 target sites were chosen using the CRISPR online design tool CHOPCHOP (http://chopchop.cbu.uib.no). Oligos with the single-guide (sg)RNA sequences used to generate the sgRNA-expressing plasmids (using method described in [37]) are listed in S2 Table. For the knockout experiments, two sgRNA targeting sites, one around the start codon, and the other around the stop codon, of the gene, were chosen in order to completely knockout the gene of interest. To generate point mutations using the CRISPR/Cas9 system, single stranded DNA oligos (listed in S2 Table) were used as the homologous repair templates. To endogenously tag the different TSP-14 isoforms, plasmids containing the homologous repair templates were generated by following the method described by Dickinson et al [38]. Specifically, ~600-bp homology arms for each target locus were amplified from N2 genomic DNA and then inserted into the GFP^SEC^3xFlag vector pDD282 or the TagRFP^SEC^3xMyc vector pDD286 via Gibson Assembly (New England BioLabs). Hygromycin was used to select the knock-in candidate lines based on the strategy described in Dickinson et al [38].

For the single-copy insertion rescue experiments, we followed the strategy described by Pani and Goldstein [39] and chose *ttTi4348* on chromosome I as the insertion site. pAP082 was the plasmid used for the expression of Cas9 and sgRNA [39]. Repair templates were generated by replacing the ClaI and AvrII fragment in pAP088 with various fragments containing the specific promoter, cDNA or chimeric cDNA-gDNA of the specific gene, and the specific 3'UTR. All plasmids used in CRISPR/Cas9 experiments are listed in S2 Table.

## Live imaging and image analysis

L4 stage or young adult worms were mounted on 2% agarose pads with 10 mM levamisole (CAS 16595-80-5; Sigma-Aldrich). A Zeiss LSM i880 microscope with Airyscan equipped with 40× Fluar objective (N.A. 1.3) or 60× Plan-apochromat objective (N.A. 1.4) using Immersol 518F oil (Carl Zeiss) was used to capture both the fluorescent and differential interference contrast (DIC) images at super-resolution. Images were viewed and processed with ZEN software. GFP was excited at 488 nm, and RFP or TagRFP was excited at 561 nm. GFP emission was captured with the BP495-550 filter, and RFP or TagRFP emission was captured with the LP570 filter. To determine the subcellular localization of GFP::TSP-14A, and the co-localization of TSP-14A with TSP-12, quantitative colocalization analysis was performed using the JACoP plugin of the open-source Fiji software [40]. For each image, the Costes threshold regression was used as the reference to establish a threshold, and three randomly selected square regions were chosen for each image pair. For each genotype, more than 10 worms were imaged and analyzed. Co-localization analysis was conducted by calculating both Pearson's correlation coefficient (PCC) and Mander's overlap coefficient (MOC) [41].

## Western blot analysis

For the western blotting experiments, 100 L4-worms were hand-picked into 20 μL ddH$_2$O, immediately flash-frozen in liquid nitrogen, and kept frozen for more than 30 minutes. Then 5 μL 5×SDS sample buffer (0.2 M Tris·HCl, pH 6.8, 20% glycerol, 10% SDS, 0.25% bromophenol blue, 10% β-mercaptoethanol) was added to each sample. The samples were then boiled at 95°C for 10 min, centrifuged at 13k rpm for 20 min, and stored at -20°C until gel electrophoresis. Proteins were separated using 7.5% Mini-PROTEAN TGX Precast Gels from Bio-Rad Laboratories, Inc., and transferred onto Immobilon-P PVDF membrane (MilliporeSigma) for 9 min under 1.3A and 25V using the Power Blotter Station (Model: PB0010, Invitrogen by Thermo Fisher Scientific). The membrane was incubated in EveryBlot Blocking Buffer (Bio-Rad Laboratories, Inc.) for 5 min at room temperature, and then incubated at 4°C overnight in primary antibodies diluted in EveryBlot Blocking Buffer. Primary antibodies used include mouse anti-FLAG IgG monoclonal antibody (Krackler Scientific; 45-F3165; diluted 1:2,000) and mouse anti-actin IgM JLA20 monoclonal antibody (Developmental Studies Hybridoma Bank; diluted 1:2,000). Secondary antibodies used included peroxidase-conjugated donkey anti-mouse IgG and peroxidase-conjugated goat anti-mouse IgM (all from Jackson ImmunoResearch; diluted 1:10,000). Enhanced chemiluminescence was detected using the Western Blotting Luminol Reagent (Santa Cruz Biotechnology; sc-2048). The Bio-Rad ChemiDoc MP imaging system was used to capture the chemiluminescence signal. The western blot experiment was repeated 3 times using different biological samples. Open-source Fiji software was used to quantify western blotting images.

## Supporting information

**S1 Fig. Information on the different mutations in *tsp-12* or *tsp-14* generated using CRISPR/Cas9.**
(PDF)

**S2 Fig. Transgenic reporters of *tsp-14*.**
(PDF)

**S3 Fig. The functionality of *tsp-14* knock-ins in the *tsp-12(ok239)* background.**
(PDF)

**S1 Table. Strains generated in this study.**
(PDF)

**S2 Table. Oligonucleotides and plasmids used in this study.**
(PDF)

**S1 Movie. Z-stacks of TSP-14A(NT) (*jj183*) in hypodermal cells.**
(AVI)

**S2 Movie. Z-stacks of TSP-14B(NT) (*jj192*) in hypodermal cells.**
(AVI)

**S3 Movie. Z-stacks of TSP-14AB(CT) (*jj219*) in hypodermal cells.**
(AVI)

## Acknowledgments

This paper is dedicated to the memory of the late Herong Shi, a dear friend and a dedicated colleague to many members of the Liu lab. We thank Andy Fire, Bob Goldstein, Barth Grant, Ariel Pani for plasmids or strains, and the rest of the Liu lab for helpful discussions and critical comments on the manuscript.

## Author Contributions

**Conceptualization:** Zhiyu Liu, Jun Liu.

**Data curation:** Zhiyu Liu, Jun Liu.

**Formal analysis:** Zhiyu Liu, Jun Liu.

**Funding acquisition:** Jun Liu.

**Investigation:** Zhiyu Liu, Herong Shi, Jun Liu.

**Project administration:** Jun Liu.

**Supervision:** Jun Liu.

**Validation:** Zhiyu Liu, Jun Liu.

**Visualization:** Zhiyu Liu, Jun Liu.

**Writing – original draft:** Zhiyu Liu, Jun Liu.

**Writing – review & editing:** Zhiyu Liu, Jun Liu.

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
