## [Decision Letter · Decision Letter 0]

13 Dec 2021

Dear Dr Liu,

Thank you very much for submitting your Research Article entitled 'The C. elegans TspanC8 tetraspanin TSP-14 exhibits isoform-specific localization and function' to PLOS Genetics.

The manuscript was fully evaluated at the editorial level and by independent peer reviewers. The reviewers appreciated the attention to an important topic but identified some concerns that we ask you address in a revised manuscript. We therefore ask you to modify the manuscript according to the review recommendations. Your revisions should address the specific points made by each reviewer.

Yours sincerely,

Iva Greenwald

Guest Editor

PLOS Genetics

Gregory P. Copenhaver

Editor-in-Chief

PLOS Genetics

This manuscript addresses interesting and important issues about tetraspanins using elegant genetic methods. The reviewers and I find significant strengths in the quality and rigor of this work, particular in the use of genome engineering to address the questions about isoforms in an endogenous context. I would be willing to give a positive recommendation on a revised manuscript that satisfactorily addresses the specific concerns raised by the two Reviewers. Please include a point-by-point response when you resubmit.

I have to admit that I initially shared Reviewer 1's hesitation about whether this study in itself represents a sufficient advance without any direct experiment(s) to connect different isoforms with different functions in trafficking of SUP-17 or DAF-4 Having pondered this question, I think that the Discussion on pp. 16-18 is a valid attempt to make such connections from the genetic data, but would be difficult for a reader who is not familiar with the intricacies of the C. elegans biology and may also be somewhat daunted by the nomenclature needed to follow the Discussion. I therefore recommend that the authors add a model figure for how the different isoforms may be contributing to different pathways or processes based on the genetic analysis to make the inferences from the genetics more accessible.

Reviewer's Responses to Questions

**Comments to the Authors:**

Reviewer #1: This study by Liu et al., describes the differential functions and localization patterns of the two TSP-14 tetraspanin isoforms in C. elegans, A, and B. Previous studies by this group and others have shown C. elegans tsp-14 and tsp-12 to be important for Notch and BMP signaling and, in the case of BMP, to regulate signaling through an intracellular trafficking mechanism. The current study is very thorough in its approach and is well written. Clearly a considerable amount of work was required to generate the required strains and the authors used a variety of complementary approaches in their tests. One strength of the article is that it provides a very good example for how to analyze multiple protein isoforms using CRISPR and transgenic approaches. A second strength is that the studies provide some insights into the specific roles and localization patterns of TSP-14 A and B isoforms and set the authors for future studies into their molecular and cellular functions. In terms of the immediate study, most of my specific comments can likely be addressed through writing together with some additional data analysis and clarifications.

A more difficult question comes down assessing to the extent to which this study advances the understanding of the system. Namely, does it represent a substantial step forward or is it currently more descriptive and incremental (and if so, does that matter)? This group has previously been shown that tsp-14 and tsp-12 are genetically redundant and have also reported expression patterns for TSP-14(A/B). The major take homes from this study are that the two TSP-14 isoforms have different patterns of localization and different functional genetic interactions with tsp-12. What was admittedly less clear is how these findings come together to form a coherent mechanism or provide a clear advance in biological insight. For example, how these isoforms might connect back to BMP trafficking. I would also note that in some cases the data were somewhat inconclusive or appeared to have some inherent contradictions (see several points below), which did weaken the findings to some extent. In any case, because this type of evaluation is subjective and journal dependent, I raise these points but leave it with the authors to make their case and with the editor to make any final judgements.

Specific comments:

Line 113. The statement that “tsp-14b has an additional 163bp segment upstream of the shared 126bp sequences” seems incorrect. tsp-14b has an additional 72bp, thereby adding 24 aa.

Line 130. Maybe state explicitly that the various Met mutations would prevent any in-frame product being made through a downstream ATG.

Figure 2. 2A. Also show the extent of jj95 deletion? 2C¬–M. These images were very hard to see, in part because the resolution was so poor. The ROI (the vulva) could be made much larger to fill most of the frame. In addition, if the images contained lethal embryos, that wasn’t clear. Although I generally accept statements that 100% or 0% of the double mutants were/weren’t Vul etc., but it would still be good to put a number on that (n>100?). Was VPC induction zero or 100% or something in between? 2P. Significance values (Fisher’s etc) should be likely be provided when making conclusions about differences in penetrance.

Figure 2 and others. In my view, it would be helpful to consistently substitute intuitive/informative names for the generated variants rather than using jj-alleles (e.g., in panel 2P). This was done to some extent (e.g., 2B), but doing so more consistently throught out the paper would make the reading and interpretation easier. For example, with expression studies it might be useful to refer to the three GFP tags as TSP-14AB(CT), TSP-14AB(NT), TSP-14B(NT) etc.

Figure 2B and others. Is the statement that the error bars are 95% CI correct? More typically box and whiskers bars represent the range (not including outliers), which is what these appear to be. Moreover, if they were indeed 95% CIs, then the reported low p-values for a number of the comparisons would not be possible it seems.

Line 140. Does EMB lethality refer to maternal effect lethality (otherwise could not score other larval and adult parameters like Susm)?

Figures 2, 3, and S2. If I understood the data correctly, it seems hard to easily rectify the results in Figures 2, 3, and S2 regarding the functions of the two isoforms. With respect to body length, Fig 2B indicates a specific role for TSP-14A with no detected function for TSP-14B. In contrast, Fig 3C indicates at least as strong a role for TSP-14B in body length as TSP-14A. With respect to Susm, in Fig 2P the tsp-12; tsp-14B double suppressed to only 37% versus 80% for tsp-12; tsp-14 (double null). But the data in Fig 3B indicate suppression of “tsp-12; tsp-14B” (jjSi388 or jjSi401) to be 80%, equal to that of the double null. Moreover, in Fig 3B the “tsp-12; tsp-14A” (jjSi390 or jjSi402) clearly suppresses to a strong extent (~50%), which is well above tsp-12 alone, in contrast to what was observed in Fig 2P. Lastly, Fig S2D indicates that the “tsp-12; tsp-14B” (Ex4848) and the “tsp-12; tsp-14A” (Ex4907) are either very weakly or not suppressed. Because extrachromosomal arrays were used in this experiment, it seems possible that over- or mis-expression of the isoforms could account for the observed rescue. On the one hand I give credit to the researchers for being thorough and carrying out these parallel approaches. And it’s potentially interesting that they got different results. But it also seems difficult to make any clean conclusions and some of these discrepancies weren’t directly addressed.

Figure 4. 4D. Though intuitive, it might be helpful to have arrows marking each isoform. 4C. The Susm effect of tsp-14(jj183) is very minor – is it statistically significant? Although the logic for these experiments were clear enough, the overall findings in this figure (the tags don’t strongly affect function) seem quite minor and more in line with a supplemental-type figure.

Figure 5 and associated text. The DIC panels for A’, C’, F’, G’, and K’ didn’t show up on my PDF for some reason. I also did not see any 5N. Regarding the statement that “TSP-14A is primarily localized within intracellular vesicles, while TSP-14B is mainly localized on the cell surface,” it may be hard for most readers to recognize this from the images. Perhaps providing a z-stack location in the panel or supplemental z-stack movies that scan through the worm, would helpful. With respect to differences in the intensity of GFP expression, this might best be represented with a simple quantification of intensities.

Figure 6 and associated text. I was slightly confused about why the text referred to the images as TSP-14A. I realize that TSP-14A expression is much higher in the epidermis than TSP-14B and is somewhat less apical, but in principle the GFP-tagged strains would mark both TSP-14A and TSP-14B. Perhaps simply referring to it as “TSP-14AB(NT)” and making the above points would be most accurate? Along those lines, was there a reason not to use some of the other generated variants that would definitively mark A (or B) isoforms only? 6H. Very minor, but the Mander’s value (GFP::TSP-12 that overlaps with TagRFP::TSP-14A (the forth bar) for this seems rather high based on the representative images (E,F,G).

Figure 7 and associated text. The text leads with the statement that “TSP-14B::GFP::3xFLAG in tsp-14a null (jj304 jj319) animals is localized to the basolateral surface of vulval and hypodermal cells.” But the data in Fig 7 refer specifically to intestine and pharynx, which was something of a disconnect. Was normal TSP-14B expression detected at the BL membrane in these tissues and is there a reason not to report findings for vulval and hypodermal cells? Also, was there an a priori reason to suspect that TSP-14B localization would depend on TSP-14A?

Figure 8. Again, you may want to provide supporting statistical data when comparing suppression levels of different alleles (also mentioned above).

Reviewer #2: The manuscript submitted to PLoS Genetics by Liu et al reports the identification of the differential functions of two isoforms of TSP-14, a C. elegans tetraspanin that plays multiple important roles in development. Previously, the Liu Lab reported that TSP-14 playd redundant roles with TSP-12 in regulating body size, embryonic development, and the postembryonic development of the mesoderm and the vulva. In this report, the authors have found that the two isoforms of TSP-14, TSP-14A and TSP-14B, display differential expression pattern, different subcellular localization, and differential functions. They have further found that the 24 amino acid peptide existing in the N-terminus of TSP-14B but absent from TSP-14A carries the code that governs the specific basolateral plasma membrane localization of TSP-14B and proposed that this code is a di-leucine motif. Through dissecting the functions and regulation of the two TSP-14 isoforms, this report reveals the diversity of the functions and regulation patterns of the tetraspanin family proteins and suggests that further in vivo and in vitro investigations will lead to the discoveries of the novel molecular mechanisms behind the complex regulation pattern of tetraspanins. The knowledge of the tetraspanin family of proteins is relatively scarce. This report thus is of high significance and will attract readers of multiple fields. In general, cell biologists, developmental biologists, and molecular biologists will be interested in reading this report, in particular researchers studying Notch and TGF-beta signaling, the regulation of membrane proteins, and intracellular trafficking.

This research is carried out in the most rigorous manner in all aspects. For example, the CRISPR-Cas9 technique is used to generate knock-in and knockout lines. As a result, the GFP-tagged reporters for the functional subcellular localization studies are all expressed as a single copy reporters, ruling out the possible artifacts of overexpression from multiple copies of the transgenes. In addition, the authors generated both N- and C-terminal GFP tagged reporters for TSP-14A and TSP-14B, and through comparison of the subcellular localization pattern of the N- and C-terminally tagged reporters, ruling out possible artifact of positional specific insertion of the GFP tag.

This manuscript is clearly and concisely written. I do not have any major concerns. I have the following specific concerns which I wish the authors would address with discussion or new experiments.

Specific Concerns:

1. Page 11, “TSP-14A colocalizes with both the early endosome marker RAB-5 and the late endosome marker RAB-7, but shows little co-localization with the recycling endosome marker RAB-11 (Figure 6I-6R).”

--Given that RAB-7 is also associated with lysosomes in addition to late endosomes, the co-localization of TSP-14A and RAB-7 might indicate that TSP-14A is also enriched on lysosomes.

2. Fig 7B and Page 12 claim that TSP-14B::GFP is localized to the basolateral membrane of the intestinal cells.

--From Fig 7B, it looks that the GFP signal is strongly enriched on the luminal side of the intestine, which is supposed to be the apical side of intestinal cells. Although the basolateral membrane localized GFP is also visible, the signal intensity is weaker than that observed on the luminal side.

3. page 13, “We found that the GFP tagged AQCAA mutant TSP-14B protein in jj322 jj192 worms becomes localized to the apical side of the developing vulva (compare Figure 8C vs 8F) and in intracellular vesicles of hypodermal cells (Figure 8D vs. 8G), just like TSP-14A”.

--Related to point 2 above, after studying Figure 8 C/F and D/G, what I observe is that the AQCAA mutations of the proposed di-leucine motif of TSP-14B primarily change the GFP localization from the plasma membrane to intracellular vesicles rather than from the basolateral to apical membrane. One possible function of the EQCLL motif might be to retain TSP-14B on the plasma membrane by slowing down the entry of PM-localized TSP-14B into the endocytic recycling pathway. Is this idea testable?

4. Regarding the extra 24 amino acids of TSP-14B, which contains the di-leucine motif, is the di-leucine motif the only part that is important for the function of TSP-14B? When tsb-14b cDNA bearing the “AQCAA” mutations is expressed under the promoter that drives tsp-14a expression, given that TSB-14B(AQCAA) is intracellular localized, like TSP-14A, can it replace tsp-14a function?

5. Page 16, “Three lines of evidence suggest that TSP-14B, but not TSP-14A, is the major player that functions redundantly with TSP-12 in patterning the postembryonic mesoderm”.

--As reported here, TSP-14B is plasma membrane localized, yet TSP-12 is known to be localized to endosomes, it is hard to reason how TSP-14B functions in a molecular activity redundant to TSP-12. It is more likely that the pathway includes TSP-14B and the pathway includes TSP-12 display redundant functions.

**Have all data underlying the figures and results presented in the manuscript been provided?**

Reviewer #1: **No: **Some panels were missing

Reviewer #2: Yes

PLOS authors have the option to publish the peer review history of their article (what does this mean?). If published, this will include your full peer review and any attached files.

Reviewer #1: No

Reviewer #2: **Yes: **Zheng Zhou

---

## [Decision Letter · Decision Letter 1]

10 Jan 2022

Dear Dr Liu,

We are pleased to inform you that your manuscript entitled "The C. elegans TspanC8 tetraspanin TSP-14 exhibits isoform-specific localization and function" has been editorially accepted for publication in PLOS Genetics. Congratulations!

Yours sincerely,

Iva Greenwald

Guest Editor

PLOS Genetics

Gregory P. Copenhaver

Editor-in-Chief

PLOS Genetics

Comments from the reviewers (if applicable):

Both reviewers are satisfied with your response and strongly support publication, so I am happy to recommend acceptance.

(With Reviewer 1 tipping his/her hat to you for not following my recommendation to include a model/summary figure, I will not push the point further. The rigorous genetics you do in this paper will be a pleasure for geneticists to read, but I thought a visual representation of your Discussion points might help make your work more accessible to readers in the tetraspanin field who are principally biochemists and find their way to your paper using keywords.)

Reviewer's Responses to Questions

**Comments to the Authors:**

Reviewer #1: The authors have done a thorough job addressing the specific concerns as directed. Some of the modifications were very nice, such as the Z-stack movies. I also 'tip my hat' to their decision to not include a model that they do not have confidence in. Lastly, I noticed this time around that one of the authors was deceased, which is a horrible thing to have happened and I am genuinely sorry to hear that.

Reviewer #2: I am satisfied with the revised manuscript and the responses to reviewers. The authors address all of my concerns. The figure included in the rebuttal letter is particularly helpful in demonstrating the basal-lateral localization in intestinal cells.

**Have all data underlying the figures and results presented in the manuscript been provided?**

Reviewer #1: Yes

Reviewer #2: Yes

PLOS authors have the option to publish the peer review history of their article (what does this mean?). If published, this will include your full peer review and any attached files.

Reviewer #1: No

Reviewer #2: No

**Data Deposition**

http://datadryad.org/submit?journalID=pgenetics&manu=PGENETICS-D-21-01478R1

**Press Queries**

---

## [Editor Report · Acceptance letter]

24 Jan 2022

PGENETICS-D-21-01478R1 

The *C. elegans* TspanC8 tetraspanin TSP-14 exhibits isoform-specific localization and function 

Dear Dr Liu, 

We are pleased to inform you that your manuscript entitled "The *C. elegans* TspanC8 tetraspanin TSP-14 exhibits isoform-specific localization and function" has been formally accepted for publication in PLOS Genetics! Your manuscript is now with our production department and you will be notified of the publication date in due course.

With kind regards,

Agnes Pap

PLOS Genetics

On behalf of:
